

# Clinical characteristics of pathogens in children with community-acquired pneumonia were analyzed *via* targeted next-generation sequencing detection

Junhua Zhao, Mingfeng Xu, Zheng Tian and Yu Wang

Pediatrics Department, Xiantao Maternity and Child Healthcare Hospital, Xiantao, China

## ABSTRACT

**Background**. The primary purpose of this study was to detect the pathogen species using targeted next-generation sequencing (tNGS) to investigate the characteristics of community-acquired pneumonia (CAP)-related pathogens in children in Xiantao city, Hubei province, China.

**Methods**. A total of 1,527 children with CAP were prospectively recruited from our hospital between May 2022 and February 2023. Information on age and sex was collected from the medical records. Pathogen detection was performed using standard detection methods and tNGS.

**Results**. The positive coincidence rate of standard detection methods and tNGS were 61.95% (946/1,527) and 97.05% (1,482/1,527), respectively. Among the 1,482 children with CAP, the numbers of bacteria, virus, chlamydia, and mycoplasma infection were 1,188, 975, 321, and 1, respectively. Co-existing species showed high prevalence in CAP, and the prevalence of children infected with only one pathogen was 20.31%. The numbers of children infected with two and three pathogens were the highest, accounting for 29.22% and 25.17%, respectively. Among the 44 pathogens detected using tNGS, 17 species of bacteria, 25 species of viruses, one species of chlamydia, and one species of mycoplasma were documented. Among all infectious pathogens, the top five were *Haemophilus influenzae*, *Acinetobacter baumannii*, *Streptococcus pneumoniae*, human herpes virus type 5 (HHV-5), and *Mycoplasma pneumoniae*. The results showed that pathogenic infections in children with CAP were related to age but not to gender.

**Conclusion**. The infection pathogens in children with CAP were complex and the incidence of co-existence was observed to be high. The pathogens involved in CAP were closely related to the age of the child. In addition, tNGS was shown to better identify pathogens than the standard detection method, which is crucial for improving the accuracy of early CAP diagnosis and initiating appropriate treatment in a timely manner, ultimately enhancing treatment outcomes.

Corresponding author
Junhua Zhao, 13507226238@163.com

## INTRODUCTION

Respiratory infections are the primary cause of admission to pediatric intensive care units (ICU) and death, therefore placing a significant burden on families and the social healthcare service system worldwide (*Du et al., 2023*; *El Bcheraoui et al., 2018*; *Ye et al., 2023*). Diseases of the respiratory system can lead to infant mortality, especially acute respiratory infections such as community-acquired pneumonia (CAP) (*Mebrahtom, Worku & Gage, 2022*). The pathogens that cause CAP are widespread and diverse and include bacteria, viruses, chlamydia, and mycoplasma (*Liu et al., 2023*). In clinical practice, bacterial, fungal, viral, and mycoplasma infections often occur simultaneously, making it difficult to identify the specific pathogens (*Cillóniz et al., 2011*). The gold standard for diagnosing infectious pathogens in clinical settings has poor sensitivity and is too slow to guide early and targeted antimicrobial therapy (*Zhang, Yang & Makam, 2019*). Owing to the similarity in infection symptoms among different pathogens, doctors typically initiate broad-spectrum antibacterial treatment in the early stages, which often result in antibiotic abuse and increased drug-resistant bacteria. Therefore, rapid and accurate identification of pathogens can improve the detection efficiency of infectious diseases and avoid the overuse of antibiotics.

Recently, quantitative polymerase chain reaction (qPCR) has been used to detect respiratory viruses in children. Compared to standard detection methods, qPCR-based rapid pathogen arrays, such as the TaqMan Array Card and nested multiplex PCR, can promptly detect infected pathogens and increase the number of species identified through microbial detection, thus reducing the use of inappropriate antibacterial therapy, which is important for infection control and treatment (*Cavallazzi & Ramirez, 2018*; *Clark et al., 2023*; *Yen et al., 2023*). Combined with standard detection methods, qPCR can increase the positivity rate of CAP pathogen detection to 80% (*Yen et al., 2023*). However, the number of pathogenic species that can be detected simultaneously using qPCR are limited. Another rapid detection method is metagenomic next-generation sequencing (mNGS), which can detect and analyze all potential pathogens in a sample (*Shahrajabian Hesam & Sun, 2023*). Compared to standard detection methods, mNGS can identify more species of infectious pathogens, co-existence pathogens overlooked when using standard detection, and pathogenic pathogens beyond the scope of standard detection in bronchoalveolar lavage fluid (BALF) samples from children with CAP. In addition, mNGS can identify pathogens that are resistant to antibiotics (*Chen et al., 2023*; *Gao et al., 2022*; *Yang et al., 2022*). Overall, mNGS detection can improve the sensitivity of pathogen detection and yield results from oropharyngeal swab samples from patients with severe non-responding pneumonia within 24 h (*Wang et al., 2020*). However, mNGS has the following limitations in clinical applications: its high sensitivity leads to false positives, relatively high cost, inability to fully detect drug resistance, long testing period, and subjective interpretation of results (*Charalampous et al., 2019*; *Zheng et al., 2021*). In addition, mNGS detects host and pathogen gene profiles simultaneously, resulting in a large amount of mNGS detection data that requires experienced data analysts to assess (*Guo et al., 2022*).

Targeted next-generation sequencing (tNGS) detection can purposefully perform parallel sequencing of thousands of short DNA sequences in a single test and is a low-cost method that can detect multiple target genes with a minimum amount of DNA. tNGS has the advantages of a short processing time, multiple detection species (typically > 200), and less generated and processed data (*Fontanges et al., 2016*). In addition, tNGS has a high detection accuracy for central nervous system infections, drug-resistant *Mycobacterium tuberculosis*, and hearing loss variant gene screening in children undergoing neurosurgery (*Li et al., 2023*; *Luo et al., 2022*; *Wu et al., 2022*). tNGS is also used to detect pathogens in adults with pneumonia and respiratory pathogens in bronchoalveolar lavage fluid samples, and exhibits high sensitivity and specificity (*Gaston et al., 2022*; *Li et al., 2022*). tNGS detection combines the advantages of qPCR and mNGS and compensates for their disadvantages. First, the detection target of tNGS only includes preset pathogens; therefore, it is not affected by human host genes or pathogens from other sources. Second, this method can be used to identify the subtypes and drug resistance of pathogens. Third, tNGS can achieve true quantitative detection of pathogens by obtaining the accurate number of copies of each detected pathogen in the sample. Finally, the amount of sequencing data for tNGS samples is approximately 1,000-fold that of mNGS; thus, so the cost of tNGS detection is low and the detection cycle is short compared with that of other methods (*Fontanges et al., 2016*). However, the usefulness of tNGS for detecting the pathogens associated with CAP in infants and young children remains unclear.

The primary purpose of this study was to analyze and compare the effectiveness of standard and tNGS diagnoses and investigate the epidemiological characteristics and mixed infection characteristics of CAP-related pathogens in children in Hubei province, China. First, tNGS was performed to detect pathogens related to CAP in deep sputum samples from children, then the effectiveness of standard and tNGS diagnoses was analyzed. Next, the epidemiological and mixed characteristics of CAP-related pathogenic infection in children were analyzed based on the results. In addition, the relationship between the characteristics of CAP-related pathogens and age and gender were analyzed.

## MATERIALS AND METHODS

### Study design

This is a prospective cohort study in which the similarities and differences between tNGS and standard detection methods of pathogens were analyzed and the primary infectious pathogens in children with CAP in our hospital were clarified. Written informed consent was obtained from the legal guardians of all participants. This study was approved by the Ethics Committee of Xiantao Maternity and Child Healthcare Hospital (2022/04/09). The children were prospectively recruited from our hospital between May 2022 and February 2023. The inclusion criteria were as follows: (1) all patients were diagnosed with CAP, and (2) their ages ranged from 1 month to 18 years. The diagnostic criteria for CAP is based on the Guidelines for the Management of Community-Acquired Pneumonia in Children (Revised in 2013) (Part I). Based on these guidelines, the following conditions should be present: (1) respiratory symptoms such as fever and cough; (2) primary signs

include increased respiratory rate, respiratory distress, rales, or bronchial breath soundin the lungs, as well as other respiratory manifestations; and (3) radiographic findings may show parenchymal or interstitial abnormalities in the lungs, with or without the occurrence of pulmonary complications. The exclusion criteria were as follows: (1) patients with CAP who have also been diagnosed with tuberculosis, lung tumors, or other lung diseases; (2) patients who received antibiotic treatment before testing; (3) those with incomplete clinical data; (4) those with pneumonia that occurred after 48 h of hospitalization; and (5) disagreement with tNGS detection. A total of 1,527 patients were included is this study. Patient information, including age and gender, was collected from the medical records.

## Deep sputum collection

Sterile collection containers and water were prepared, and guidance was provided to the patient for collecting morning sputum for examination. Generally, the natural coughing method is used to collect sputum. In brief, the mouth was repeatedly rinsed with water to remove foreign objects before collecting sputum, then two to three deep breaths were taken and the first and second mouthfuls of sputum were forcefully coughed out from the trachea into a sterile collection container. For patients who experience difficulty coughing up sputum, nebulized steam inhalation of 3–5% sodium chloride solution was used to help induce sputum production. For patients with little or no sputum, bronchoscopy or tracheal puncture was used to obtain sputum from the trachea. When sputum collection was difficult for young children, a disinfection cotton swab was used to stimulate the throat and induce cough reflex, and the collected sputum specimen was obtained using the throat swab. The sputum was evaluated through microscopy.

## Clinical composite diagnosis

Pathogen diagnosis requires a comprehensive judgment of clinical symptoms, laboratory examination, imaging, and standard methods (including bacterial culture, antibody tests, and PCR detection). The specific types of standard methods are listed in Table 1.

## tNGS testing

The tNGS sequencing reaction general kit (Kingcreate, Guangzhou, China) can detect 71 pathogens (Table 1) and four drug-resistant mutation sites of the 23S rRNA gene of *M. pneumoniae*: A2063G, A2064G, A2067G, and C2617G. The detection steps were as follows: Briefly, deep sputum samples were digested at 25 °C for 30 min using sputum digestion solution. Next, 1.3 mL of eluent was processed through high-speed centrifugation (4,000×g, 5 min) to enrich pathogens. The supernatant was removed and retained until a 250-μL sample was obtained. Nucleic acid extraction was performed automatically using this sample and nucleic acid extraction reagent (IVD5412-F-96; Guangzhou Magen Biotechnology Co., Ltd., Guangzhou, China) with an automated extraction workstation (KingFisher flex; Thermo Fisher Scientific, Waltham, MA, USA). Nuclease free water (250 μL) was used as the quality control. The library was prepared according to the manufacturer's instructions (Multiple Joint Detection Kit for Pathogenic Microorganisms, Kingcreate, Guangzhou, China). The target pathogens were detected using the tNGS sequencing reaction general kit, according to the manufacturer's instructions, and a Gene

**Table 1  Seventy-one pathogens detected by tNGS.**

| | | |
|---|---|---|
| Bacteria (24) | *Corynebacterium diphtheriae* | Rhinovirus |
| | *Bordetella pertussis* | Rhinovirus type A |
| | *Acinetobacter baumannii* | Rhinovirus type B |
| | *Klebsiella pneumoniae* | Rhinovirus type C |
| | *Streptococcus pneumoniae* | Influenza C virus |
| | *Enterococcus faecalis* | Enterovirus |
| | *Streptococcus pyogenes* | Enterovirus Group 71 |
| | *Fusobacterium necrophorum* | Enterovirus Group A |
| | *Staphylococcus aureus* | Enterovirus Group B |
| | Katamorella | Enterovirus Group C |
| | *Neisseria gonorrhoeae* | Enterovirus Group D |
| | *Haemophilus influenzae* | Rubella virus |
| | *Neisseria meningitidis* | Influenza A virus |
| | *Arcanobacterium haemolyticum* | Influenza A virus H1N1 |
| | *Enterococcus faecium* | Influenza A virus H1N12009 |
| | *Legionella pneumophila* | RNA viruses ($n = 32$) · Influenza A virus H3N2 |
| | *Stenotrophomonas maltophilia* | Influenza A virus H5N1 |
| | *Streptococcus dysgalactiae* | Influenza A virus H7N9 |
| | *Pseudomonas aeruginosa* | Measles virus |
| | *Streptococcus agalactiae* | Human parainfluenza virus type 1 |
| | *Streptococcus constellatus* | Human parainfluenza virus type 2 |
| | *Streptococcus pharyngitis* | Human parainfluenza virus type 3 |
| | *Serratia marcescens* | Human parainfluenza virus type 4 |
| | *Streptococcus intermedius* | Human respiratory syncytial virus type A |
| DNA viruses ($n = 11$) | Human herpesvirus type 1 | Human respiratory syncytial virus type B |
| | Human herpesvirus type 2 | Human coronavirus 229E |
| | Human herpesvirus type 3 | Human coronavirus HKU1 |
| | Human herpesvirus type 4 | Human coronavirus NL63 |
| | Human herpesvirus type 5 | Human coronavirus OC43 |
| | Human Boca virus type 1 | Human metapneumovirus |
| | Human Parvovirus B19 | Mumps virus |
| | Human Adenovirus | Influenza B virus |
| | Human Adenovirus Group B | Chlamydia ($n = 3$) · Chlamydia trachomatis |
| | Human Adenovirus Group C | Chlamydia psittaci |
| | Human Adenovirus Group E | Chlamydia pneumoniae |
| Mycoplasma ($n = 1$) | *Mycoplasma pneumoniae* | |

Sequencer (Model: KM MiniSeqDx-CN; Kingcreate). The Pathogenic Microbial Data Analysis and Management System (v1.0) was used for automatic data analysis (Kingcreate). The data quality requirements were as follows: $Q30 \geq 75\%$, minimum original reads $\geq$ 50k, and internal reference gene amplification reads $\geq$ 200.

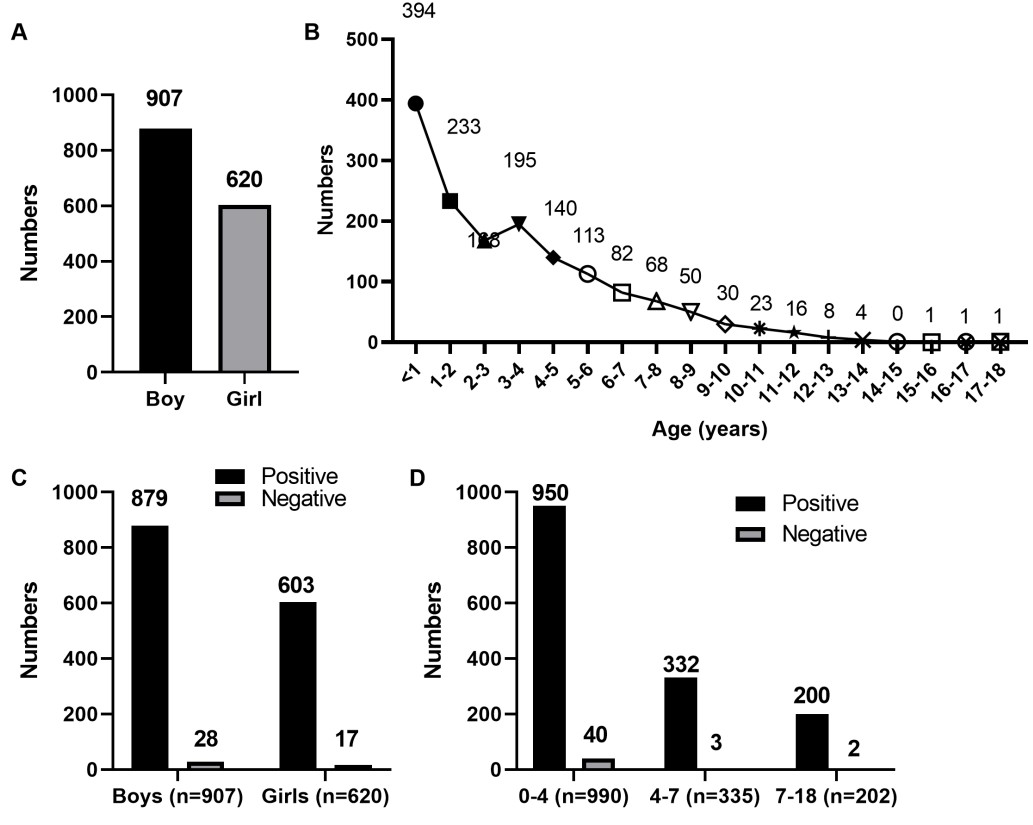

**Figure 1** **Age distribution of 1,527 children with CAP.** (A) Sex distribution of 1,527 children with CAP. (B) Age distribution of 1,527 children with CAP. (C) Sex distribution of 1,527 children with CAP after tNGS detection. (D) Age distribution of 1,527 children with CAP after tNGS detection.

## Statistical analysis

Data analysis was performed using the SPSS software (v.19.0; IBM SPSS Inc., Chicago, IL, USA). Statistical significance was set at $P < 0.05$, and two-tailed tests were used for hypotheses. Frequency counts and percentages were used to present the results. Categorical data were analyzed using the chi-square test or Fisher's exact test (frequency, <5) method to examine the differences between the detection results of tNGS and standard detection methods. Consistency between tNGS and standard detection was analyzed using the kappa test. Differences between the diagnostic efficiency of tNGS and standard detection was analyzed using the paired McNemar chi-square test.

## RESULTS

### Clinical characteristics

A total of 1,527 children with CAP (907 boys and 620 girls; Fig. 1A) were enrolled in this study. The age distribution of the patients is shown in Fig. 1B.

**Table 2   Comparison of consistency between tNGS detection and standard detection.**

| tNGS detection | Standard detection | | Total |
|---|---|---|---|
| | **Positive** | **Negative** | |
| Positive | 941 | 541 | 1,482 |
| Negative | 5 | 40 | 45 |
| Total | 946 | 581 | 1,527 |

## Differences between tNGS and standard detection methods

Using the final clinical diagnosis as a reference, we evaluated the performance of tNGS and standard detection methods for pathogen identification. Based on the clinical results, the positive coincidence rates of the standard detection and tNGS methods were 61.95% (946/1,527) and 97.05% (1,482/1,527), respectively. The comparison of consistency between tNGS and standard detection is shown in Table 2. The results showed that the consistency between the two detection methods is poor (Kappa = 0.077). Positive and negative coincidence rates of sex and age distribution in the 1,527 children with CAP after tNGS detection are shown in Fig. 1C and 1D. Therefore, the clinical detection effect of tNGS was better than that of the standard detection methods ($P < 0.001$). The standard detection method and tNGS identified 11 and 44 pathogens, respectively. Standard detection methods only identified three instances of pathogen co-existence, whereas tNGS detected eight such instances.

## Characteristics of pathogenic infections

The species of pathogens that co-existed in the 1,482 children detected through tNGS are presented in Fig. 2. Among the 1,482 children with CAP, 1,188 had bacterial infection, 975 had viral infection, 321 had chlamydia, and one had chlamydia infection (Fig. 2A). The number of children infected with only one pathogen was 301 (20.31%). The number of children infected with two pathogens was the highest at 433(29.22%), followed by children infected with three pathogens at 373(25.17%) (Fig. 2B). In addition, 570 (38.46%) children with CAP were infected only with mycoplasma (82), Chlamydia (1), bacteria (331), or viruses (156) (Fig. 2C). Among children with CAP who were infected with only bacteria, children with only one bacterial species present were 113 (34.14%) (Fig. 2D). Among children with CAP who were infected with only viruses, children with only one viral species present were 106 (67.95%) (Fig. 2E).

## Pathogenis species

Among the 44 pathogens which had identified by tNGS, there were 17 species of bacteria, 25 species of viruses, one species of chlamydia, and one species of mycoplasma. The most common bacteria were *H. influenzae* (419/1,482, 28.27%), *A. baumannii* (403/1,482, 27.19%), and *S. pneumoniae* (399/1,482, 26.92%) (Fig. 3A). The most common viruses (including DNA and RNA viruses) were human herpes virus type 5 (HHV-5, 343/1,482, 23.14%), *human rhinovirus* type A (HRV-A, 132/1,482, 8.91%), and *Influenza A virus* (IFV-A, 129/1,482, 8.70%) (Figs. 3B–3C). In addition, the infected children included *M. pneumoniae* (321/1,482, 21.66%) infection and *Chlamydia trachomatis* (1/1,482, 0.07%)

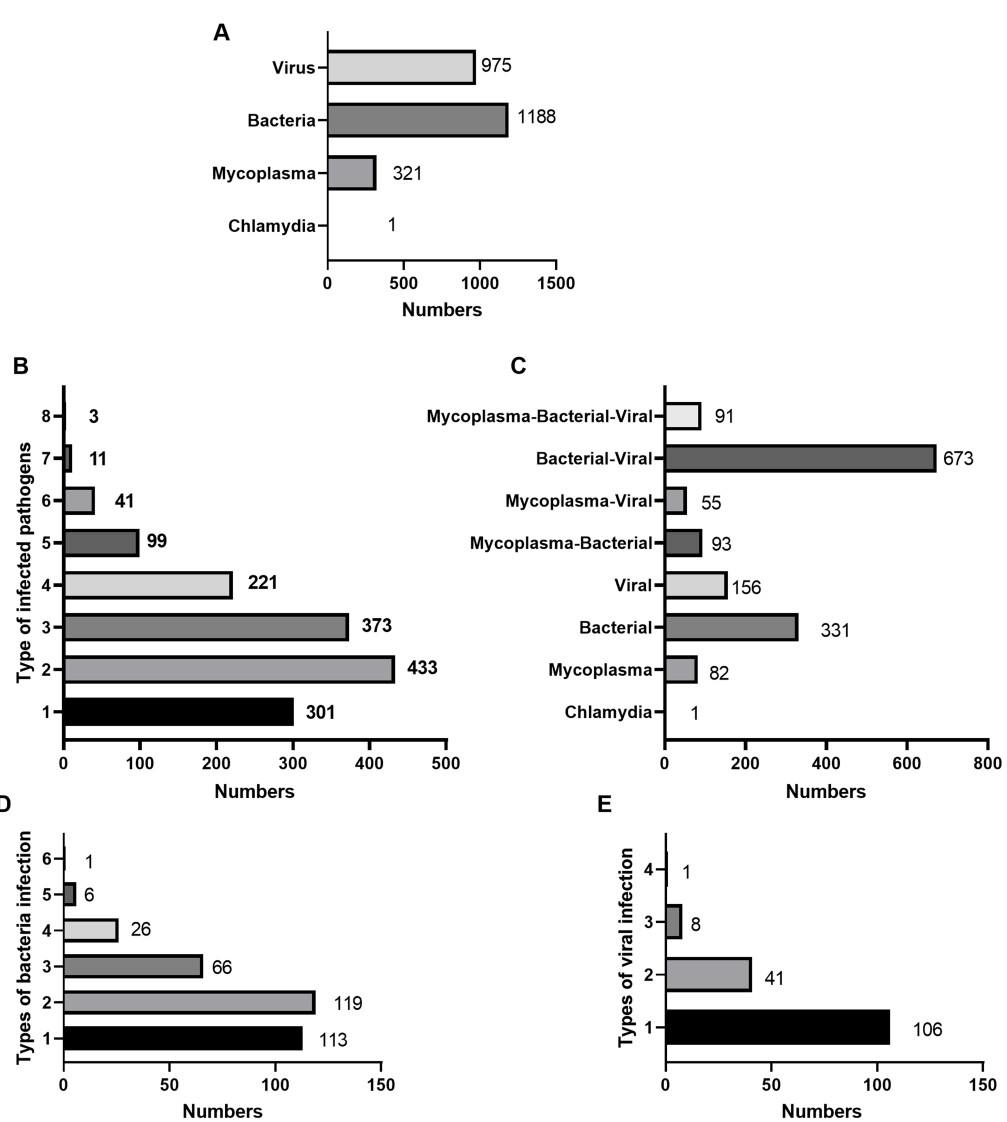

**Figure 2 Infection distribution among 1,482 children infected with pathogens.** (A) The distribution of bacteria, viruses, chlamydia, and mycoplasma. (B) The distribution of the types of co-infected pathogens. (C) The distribution of the co-infected mycoplasma-bacterial-viruses. (D) The distribution of bacteria infected types in 331 children with only bacterial infections was shown. (E) The distribution of infected-viral types in 156 children with only viral infections was shown.

infection (Fig. 3D). Among the 321 children with *M. pneumoniae* infections, 302 had drug-resistant *M. pneumoniae* infections. The top five infectious pathogens were *H. influenza*, *A. baumannii*, *S. pneumoniae*, HHV-5, and *M. pneumoniae* (Fig. 3).

## Differences in pathogenic infection between different genders and age groups

Next, the 1,482 children were divided into male and female groups. No significant differences in the infection rates of bacteria, mycoplasma, chlamydia, or viruses were

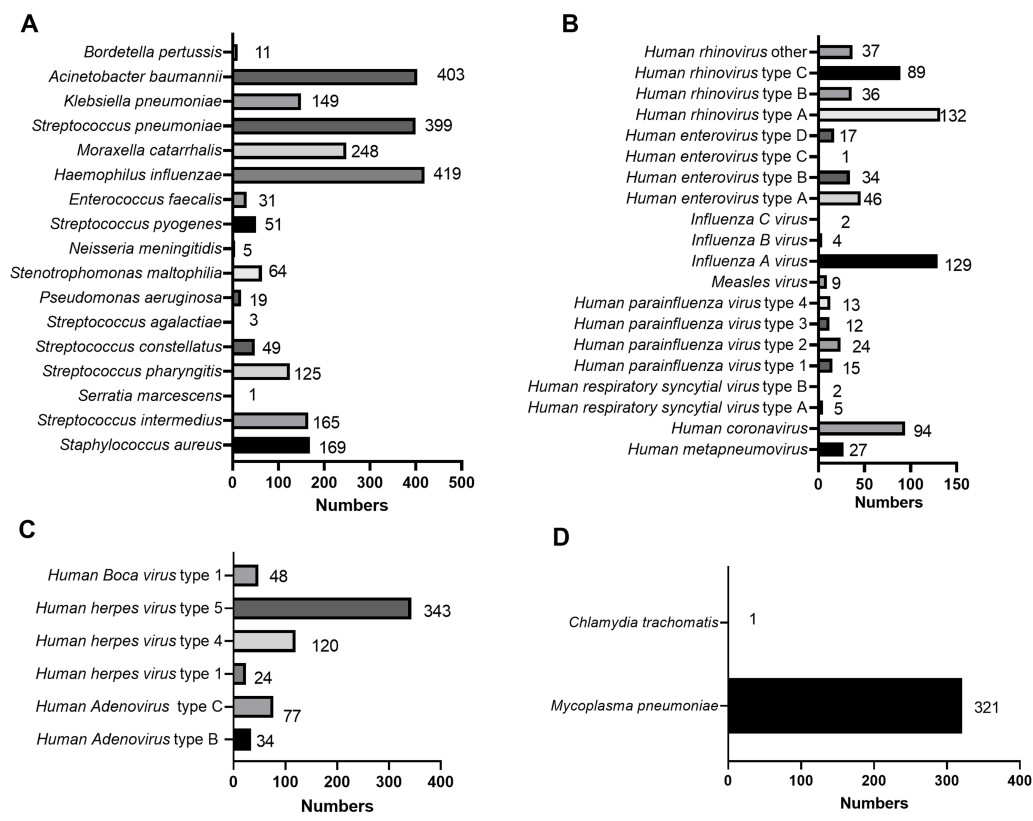

**Figure 3  Specific types of infectious pathogens in 1,482 children with CAP.** The distribution of specific types in bacteria (A), viruses (B), chlamydia (C), and mycoplasma (D) were shown.

observed between the two groups (Fig. 4A). No significant differences in the rate of pathogen co-existence were noted between the genders (Fig. 4B). In addition, no significant differences in the co-existence rates of bacteria, viruses, chlamydia, and mycoplasma were observed between the genders (Fig. 4C). Furthermore, the 1,482 children were divided into three groups: 0–4, 4–7, and 7–18-year-old groups. The viral infection rate was higher in the 0–4-year-old group than in the other two groups ($P < 0.001$), whereas the infection rates of mycoplasma/chlamydia were higher in the 7–18-year-old group than in the other two groups ($P = 0.011$). The bacterial infection rate in the 4–7-year-old group was higher than that in in the other two groups ($P < 0.001$) (Fig. 4A). The infection rate of one pathogen was the highest in the 0–4-year-old group, that of two to five pathogens was the highest in the 4–5-year-old group, and that of four to six pathogens was the highest in the 7–18-year-old group (Fig. 4B). The highest rates of single-virus infection and bacterial-virus co-existence were found in the 0–4-year-old group, the highest rates of bacterial-virus-mycoplasma co-existence were observed in the 4–6-year-old group, and the highest rates of mycoplasma infection and mycoplasma-bacterial co-existence were found in the 7–18-year-old group (Fig. 4C). However, no significant difference was observed among the three groups of bacterial infection alone and co-existence with the chlamydia virus (Fig. 4C).

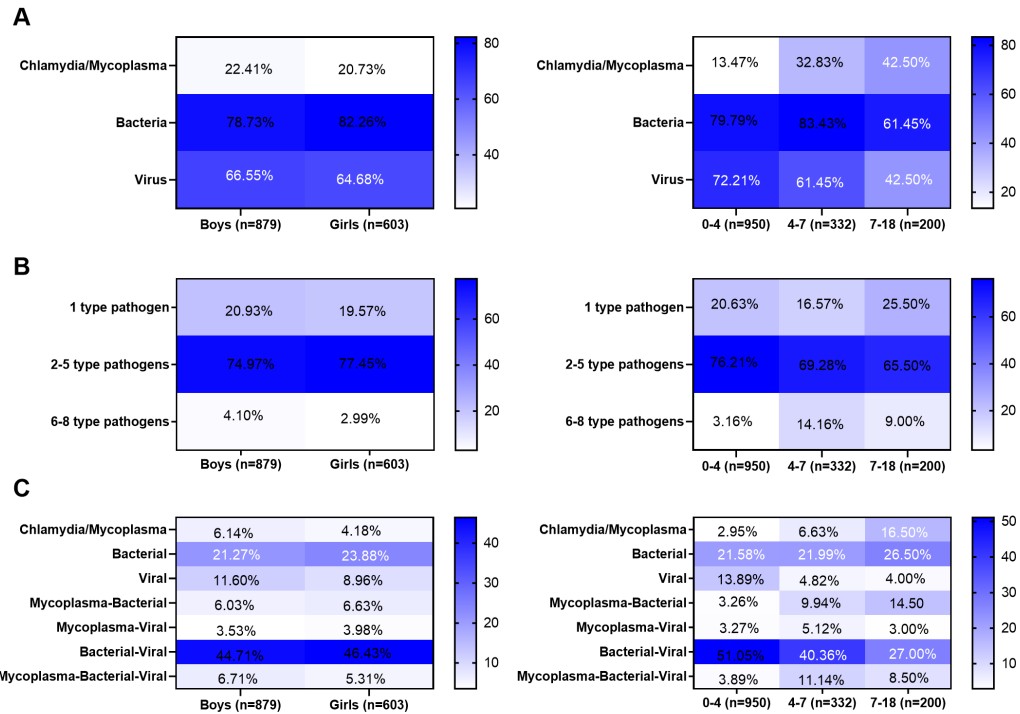

**Figure 4  Differences in pathogen infection between different genders or age groups.** (A) Infection rates of bacteria, Mycoplasma, Chlamydia, and virus between different genders or age groups. (B) The co-existence rates of pathogens between different genders or age groups. (C) The co-existence rates of bacteria-viruses-chlamydia/mycoplasma between different genders or age groups. Categorical data were analyzed by chi-square test or Fisher's exact test (when the frequency is less than 5).

## DISCUSSION

Complex infections by pathogens have synergistic effects that exacerbate the severity of CAP in children (*Yun et al., 2019*). Therefore, a novel detection method that can sensitively and specifically distinguish between the species of pathogenic infections is required to treat pediatric CAP in a timely and targeted manner. In the present study, tNGS was used to detect CAP related pathogens and the results suggested that the positive coincidence rates and the species of pathogens detected simultaneously of tNGS were higher than that of standard detection methods.

We analyzed the species of pathogens in children with CAP. The results showed that bacterial infection was the most common, followed by viral and mycoplasma infections, whereas chlamydia infection was the least common. Further results indicated that mixed bacterial and viral infections were the most common in children with pneumonia, followed by simple bacterial and viral infections. In children with CAP, mixed infections of bacteria and mycoplasma, as well as viral and mycoplasma infections, were also present, with 6.96% having mixed infections of bacteria, viruses, and mycoplasma. Compared to adults with CAP, children with CAP have more mixed viral or bacterial pathogens. Previous studies have shown that the coinfection rate of viruses and bacteria is as high as 68%,

which is an inducing factor of CAP and a risk factor for ICU admission, disease severity, and mortality (*Cawcutt & Kalil, 2017*). Viral and bacterial co-existences often lead to worse prognoses than bacterial or viral infections alone (*Kim & Kim, 2022*). This study also showed that the bacteria and viral co-existences reached 44.07%, not including the bacteria, viruses, and mycoplasma co-existence. Furthermore, we analyzed the species of infectious pathogens and determined that 77.34% were infected with two or more pathogens. Some children with CAP, even those infected with bacteria or viruses, often experience co-existence with several cells or viruses. In addition, in children with CAP infected solely with bacteria and viruses, bacterial infections usually involve multiple bacteria simultaneously, whereas viruses typically exist as a single species. These complex modes of infection seriously affect the timely diagnosis and treatment of CAP in children. Particularly, the co-infection phenomenon observed in this study suggests that clinicians should conduct more comprehensive assessments and interventions for complex cases in order to reduce the incidence of severe illness.

*S. pneumoniae, H. parainfluenzae, Staphylococcus aureus, Pseudomonas aeruginosa,* and *Klebsiella pneumoniae* are common bacteria in children with severe CAP (*McIntosh, 2002*). In addition, *S. pneumoniae, H. influenzae, Streptococcus pyogenes, Staphylococcus aureus,* and *Moraxella catarrhalis* are common bacteria that cause CAP in children younger than five years (*Leung, Wong & Hon, 2018*). *S. pneumoniae* accounts for approximately 25.00% of CAP cases in children (*Yang et al., 2022*). In addition, *S. pneumoniae* can be detected in the sputum of CAP patients (*Miyazaki et al., 2024*). The most common bacteria, *H. influenza* (28.27%), *A. baumannii* (27.19%), and *S. pneumoniae* (26.92%), were detected in children with CAP. Respiratory syncytial virus (RSV), human parainfluenza virus (HPIVs), IFV-A/IFV-B, HRV, human metapneumovirus (HMVP), and human bocavirus (HBoVs) were the most common viruses in children with CAP (*Leung, Wong & Hon, 2018*; *Pratt et al., 2022*; *Wetzke et al., 2023*). In contrast, the proportions of RSV, HPIVs, HMVP, and HBoV were 0.46%, 4.19%, 1.77%, and 3.14%, respectively. The most common viruses were HHV-5 (22.46%), HRV-A (8.64%), and IFV-A (8.45%). These results indicate regional differences in viral infection species. Therefore, diagnosis and treatment should be based on the characteristics of the area during the treatment process. In addition, chlamydia and mycoplasmas are associated with CAP in children. *M. pneumoniae* and *C. trachomatis* were the most common chlamydia and species of mycoplasma, respectively, in children with CAP. The infection rate of *M. pneumoniae* is 26.4%, which can cause longer periods of fever and cough than bacterial or viral infections (*Kuo et al., 2022*; *Yi et al., 2022*). In this study, the infected children included 321 cases of *M. pneumoniae* infection and one *C. trachomatis* infection. Notably, the proportion of drug-resistant *M. pneumoniae* was 94.08% in *M. pneumoniae* infection. These findings provide important information for clinicians, helping them develop more personalized strategies for diagnosing and treating children CAP, thereby improving treatment outcomes and patient prognosis.

The type of infectious pathogen is closely related to the age of the patients with CAP. Infection patterns and species of pathogens differ between patients with severe and non-severe CAP in an age-dependent manner (*Liu et al., 2023*). Age is a primary factor in predicting viral infection in children with CAP (*Chang et al., 2023*). *M. pneumoniae*

infection is related to age, mainly affecting children over five years old, and is age-dependent (*Smyrnaios et al., 2023*; *Yi et al., 2022*). In this study, the most common age range for viral infection, bacterial infection, and mycoplasma infection was before the age of 4 years, 4–7 years old, and 7 and 18 years old, respectively. The infection rate of one pathogen was highest in children younger than 4 years of age, whereas the infection rate of more than one pathogen was highest after the age of three. The reason may be that the children had not attended school before the age of 3 years, and the activity environment was primarily at home, with a single pathogen; after the age of three years, the main environment for children to engage in activities was school, with complex personnel and pathogens, so they may have been infected with more than one pathogen. Our study also suggests that the type of infectious pathogen is closely related to the age of the patients with CAP, similar with previous study (*Deng et al., 2023*). Therefore, combining the age characteristics of pathogenic infections can help clinicians develop more personalized treatment plans, thereby improving the diagnosis and treatment outcomes of children CAP.

This study has several limitations. This is a single center study and cannot fully represent the characteristics of CAP pathogenic infection in the entire Hubei Province. Second, there are no data comparing performance of tNGS and other detection platforms including mNGS and qPCR. Owing to the small number of children with CAP after the age of 12, children aged 7–18 were not further divided into the aged 7–12 (primary grades) group and the aged 12–18 (the high junior middle school) group. The study did not include a healthy control group for tNGS testing, so the specificity of the method remains unclear. *A. baumannii* is a Gram-negative, aerobic rod that primarily causes hospital-acquired pneumonia. However, its association with CAP is rare (*Garnacho-Montero & Timsit, 2019*). In this study, the detection rate of *A. baumannii* in CAP cases was found to be 27.19%, so whether false positives exist remains to be further investigated. In summary, improving the sensitivity and specificity of tNGS testing requires further investigation.

## CONCLUSION

The etiology of pediatric CAP can be complex, as multiple bacteria, viruses, or other pathogens as well as age may play a role. In cases where a wide variety of pathogens are present or when they are difficult to identify using conventional detection methods, tNGS technology can simultaneously detect multiple pathogens and allows for early identification of potential pathogens, thereby aiding clinicians in identifying mixed infections. This helps to avoid overlooking potential pathogenic microorganisms, particularly those that are highly pathogenic or have low colonization rates. This is crucial for improving the accuracy of early diagnosis and initiating appropriate treatment in a timely manner, ultimately enhancing treatment outcomes. This study provides a theoretical basis for the clinical application of tNGS. However, further studies with larger sample sizes are required to further assess the accuracy of tNGS detection.

## ACKNOWLEDGEMENTS

The authors express gratitude to colleagues in the Laboratory Department for Microbiological Tests for assistance during this study.

### Funding

The authors received no funding for this work.

### Competing Interests

The authors declare there are no competing interests.

### Author Contributions

- Junhua Zhao conceived and designed the experiments, performed the experiments, analyzed the data, prepared figures and/or tables, and approved the final draft.
- Mingfeng Xu conceived and designed the experiments, performed the experiments, analyzed the data, prepared figures and/or tables, authored or reviewed drafts of the article, and approved the final draft.
- Zheng Tian performed the experiments, analyzed the data, authored or reviewed drafts of the article, and approved the final draft.
- Yu Wang analyzed the data, authored or reviewed drafts of the article, and approved the final draft.

### Human Ethics

The following information was supplied relating to ethical approvals (i.e., approving body and any reference numbers):

This study was approved by the Ethics Committee of Xiantao Maternity and Child Healthcare Hospital.

### Data Availability

The raw data are available in the Supplementary File.

### Supplemental Information

Supplemental information for this article can be found online at http://dx.doi.org/10.7717/peerj.18810#supplemental-information.

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
