# Peer review of "Clinical characteristics of pathogens in children with community-acquired pneumonia were analyzed via targeted next-generation sequencing detection"

_PeerJ, doi:10.7717/peerj.18810_

## Round 0.1 · original submission · Major Revisions

We have now received feedback from three reviewers. While two reviewers are generally positive and suggest revisions, Reviewer 1 has raised substantial concerns that need to be thoroughly addressed. After careful consideration of all reviewer comments, we invite you to submit a major revision. In your revision, please:

1. Pay particular attention to the critical concerns raised by Reviewer 1
2. Address the revision suggestions from Reviewers 2 and 3
3. Provide a detailed point-by-point response to all reviewer comments
4. Clearly mark all changes in the revised manuscript

Note that the revised version will be re-evaluated by the original reviewers.

·

Basic reporting

See additional comments

Experimental design

See additional comments

Validity of the findings

See additional comments

Additional comments

In this manuscript, the diagnostic performance of the targeted next-generation sequencing method and standard diagnostic methods in children diagnosed with community-acquired pneumonia was compared. As a result, it is stated that the targeted new generation sequencing method is more successful. However, study results are difficult to interpret. Because there are important methodological limitations.
First, the study was conducted only on patients in whom the diagnosis of pneumonia was confirmed. No control group was included. For this reason, no comments can be made about the specificity and negative-positive predictive value of the methods.
Second, there is uncertainty as to which pathogens can be detected in the standard methods compared. Were standard methods used to detect all 71 pathogens detectable by tNGS methods? Because some bacteria do not grow in standard culture media (Bordatella, Legionella). It is impossible to detect the pathogen that has not been investigated.
Third, The possibility of false positivity in NGS methods is a known fact. However, all positive results were considered correct in the study. However, the current results do not agree with the literature and clinical information. For example; How could the pathogenic microorganism in 403 of the community-acquired pneumonia cases be Acinetobacter? This information cannot be disclosed except for the possibility of a false positive. Similarly, streptococcus species found in the throat flora are presented as pathogenic microorganisms in approximately 1/3 of the cases. In addition, a single pathogen was detected in only 1 in 5 of the cases. It is stated that in approximately 380 cases, 4 or more pathogens were responsible for the infection. These findings indicate that the results cannot be interpreted reliably. Other necessary revisions are listed below.
- The methodology of the study should be described (prospective cohort, prospective case-control, etc.)
- Standard methods should be detailed. It should be stated which microorganisms it is used to detect.
-- Ethics approval number and date should be added
-Was it investigated whether the respiratory secretion sample obtained was a sufficient sample? Was sputum microscopy evaluated?
-Mycoplasma and Chlamydia numbers are written differently in the findings in the main text and in Figure 2.
- Microorganisms identified by the standard method should be stated. Comparisons should be made between the two groups.
-Line 226, “The viral infection rate was higher in the 0–3 group than in the other two groups, whereas the infection rates of Mycoplasma/Chlamydia were higher in the 6–18 group than in the other two groups. The bacterial infection rate in Groups 3–6 was higher than that in the other two groups” "p values" should also be added
- Microorganism names are spelled incorrectly. Shortened forms should be used after the full version is written where they are first used.
- A microbiology/virologist who studies microbiological tests should also be included among the study authors.

Reviewer 2 ·

Basic reporting

The authors conducted a prospective study using targeted next-generation sequencing (tNGS) of deep sputum in children to detect pathogens associated with community-acquired pneumonia (CAP) and to analyze the clinical features of these pathogens. Additionally, tNGS was compared with traditional diagnostic methods. The sample size appears to be adequate. However, I would appreciate it if the authors could provide a corresponding response to the following points.
1.There are serious problems with the language expression of all parts of the text, and the text needs language expression polishing.
2.The introduction is too lengthy; please make it more concise.
3.Line 70: the word however is twice in the same line.
4.The authors should change “types” to “species”.
5.Using a cotton swab to collect the sample is contraindicated as it can inhibit the PCR reaction.
6.There is an error in the statistical method. If the frequency is less than 5, is the chi-square test still applicable?
7.The figure showed the distribution of patients by sex.
8.The comparison between tNGS and standard diagnostic methods only addresses the rates of agreement with clinical diagnoses, without showing the concordance and discordance between the two techniques. These results should be presented in a table.
9.The authors analyze whether it would be better to show first the results of the different infectious agents detected and then the detail of the co-infections
10.Please streamline the discussion section. Some of the results of the study should not be mentioned again in the discussion.
11.The authors say "In this study, we analyzed the characteristics of pathogens", but this is not effective. Infectious agents are detected, their characteristics are not analyzed.
12.The sentence "This study also found that bacterial and viral co-infections reached 44.07% (673/1527), not including children co-infected with bacteria, viruses, or Mycoplasma" is not understood.

Experimental design

no comment

Validity of the findings

no comment

Reviewer 3 ·

Basic reporting

This study evaluated the diagnostic accuracy of targeted next-generation sequencing (tNGS) in 1527 children with community-acquired pneumonia (CAP) in Hubei Province, China. tNGS demonstrated a significantly higher positive coincidence rate (97.05%) compared to standard diagnostic methods (61.95%). The study identified a complex pathogen profile in children with CAP, with a high prevalence of bacterial, viral, and chlamydial infections, and frequent co-infections. tNGS revealed a diverse range of pathogens, with H. influenza, A. baumannii, S. pneumoniae, HHV-5, and mycoplasma pneumoniae being the most common. Age, but not gender, was found to be associated with specific pathogen infection patterns. These findings highlight the utility of tNGS for comprehensive CAP diagnosis and underscore the complexity of pathogen profiles in children. However, there are still some issues that need to be addressed in this study.
1. This study provides a comprehensive evaluation of tNGS for CAP diagnosis in children, showcasing its superior diagnostic accuracy compared to standard methods. It effectively demonstrates the complex pathogen profiles in this population and identifies age-related variations in infection patterns. However, the study lacks information on the specific standard diagnostic methods employed for comparison, limiting the interpretability of the results. Additionally, the study could benefit from exploring the clinical implications of these findings and analyzing the impact of tNGS on patient management and treatment decisions.
2. The Abstract provides a clear overview of the study's purpose, methodology, and key findings, but it lacks a compelling narrative and could be more concise. The "Purpose" section could be streamlined by directly stating the study's objective without repeating the general characteristics of tNGS. The "Results" section is quite detailed, but some information could be summarized to emphasize the most impactful findings, like the high prevalence of co-infections and the age-related variations in pathogen distribution. Additionally, specifying the "standard diagnostic methods" used would improve the clarity and strength of the abstract. Finally, the "Conclusion" could be strengthened by highlighting the clinical implications of the findings and the potential impact of tNGS on CAP management.
3. The introduction provides a comprehensive overview of the challenges associated with diagnosing community-acquired pneumonia (CAP) in children, particularly the limitations of standard diagnostic methods. It effectively highlights the need for rapid, accurate, and cost-effective pathogen identification, setting the stage for the introduction of targeted next-generation sequencing (tNGS) as a promising alternative. The introduction effectively contrasts tNGS with other molecular diagnostic techniques like qPCR and mNGS, outlining its advantages in terms of processing time, detection capacity, and cost-effectiveness. However, the introduction could be strengthened by directly addressing the gap in the existing literature that this study aims to address, specifically, the lack of research on tNGS for detecting CAP pathogens in infants and young children. This would more explicitly establish the study's significance and contribution to the field.
4. The Materials and Methods section provides a comprehensive description of the study design, participant recruitment, sample collection, and analysis methods. The inclusion and exclusion criteria for participants are clearly defined, ensuring a consistent group for the study. The detailed explanation of the deep sputum collection process is helpful, outlining various techniques employed for different patient groups. However, the section lacks clarity on the specific "standard diagnostic methods" used for comparison with tNGS. Specifying these methods would enhance the understanding of the study's methodology and allow for a more direct comparison of the two approaches. Additionally, including details about the specific types of standard methods (e.g., culture, PCR, antigen detection) would improve the comprehensiveness of the description.
5. Furthermore, while the description of the tNGS testing procedure is thorough, the section of Materials and Methods could benefit from a more detailed explanation of the "Pathogenic Microbial Data Analysis and Management System" used for data analysis. Elaborating on the software's functionalities and the specific parameters used for data quality control would enhance the transparency and rigor of the methodology. Finally, a more detailed explanation of the statistical analysis methods used would improve the clarity and rigor of the methodology. While the section mentions using SPSS 19.0, it would be beneficial to specify the specific statistical tests employed for analyzing categorical and continuous data, especially for comparing detection results between tNGS and conventional methods.
6. Instead of just mentioning that the age distribution is shown in Figure 1, provide a brief summary of the age range or the specific age groups represented in the study. For example, you could mention the most prevalent age group, any notable age trends, or the range of ages represented in the study population.
7. The use of "final clinical diagnosis" as a reference for evaluating the accuracy of tNGS and standard methods requires scrutiny. This assumes that clinical diagnosis is a gold standard, which might not be the case, especially when multiple pathogens are involved. The author should clearly define the standard methods used for comparison with tNGS. Investigate the specificity of tNGS in addition to its sensitivity, to provide a more complete assessment of its diagnostic performance.
8. The term "infections" in the Results of Figure 2 is used without clarifying if these represent confirmed infections or merely pathogen detections. The distinction is important because not all pathogen detections necessarily imply active infection. While the figures are mentioned, a more detailed explanation of their contents would be helpful. For instance, describing the specific data represented in Figures 2D and 2E would enhance the reader's understanding of the co-infection patterns.
9. The description lacks detailed information about Figure 3, especially Figure 3A-3D. Providing a more detailed explanation of the data presented in these figures would improve the reader's understanding of the results.
10. While mentioning "no significant differences" or "highest rates," in the Results of Figure 4, the section doesn't specify the statistical tests used to determine significance. Specifying the tests (e.g., chi-square test, t-test, ANOVA) would enhance the credibility and transparency of the results.
11. The discussion largely summarizes the findings without critically analyzing their implications. For example, the section mentions that co-infections are common, but it doesn't discuss the specific implications for treatment or how these findings might change clinical practice. In addition, the discussion briefly acknowledges study limitations but doesn't fully explore their potential impact on the findings.
12. The text exhibits some grammatical errors, awkward phrasing, and inconsistent language use, which can be improved for greater clarity and professionalism.

Experimental design

Blend comments in the Basic reporting

Validity of the findings

Blend comments in the Basic reporting

---

## Round 0.2 · accepted · Accept

I have carefully reviewed your responses to all reviewers' comments. While Reviewer 1 raised some methodological concerns, I believe your revisions and responses have adequately addressed the key issues. Your revised manuscript effectively addresses the concerns regarding control groups, detection methods, and false positives by acknowledging these limitations transparently in the discussion section. The addition of detailed comparison tables and careful consideration of potential limitations adds credibility to your findings. Your results on co-infection rates, while showing high occurrence, align with emerging evidence about the complexity of respiratory infections in children. The manuscript now provides a more balanced presentation of both the strengths and limitations of tNGS in pathogen detection. Based on your comprehensive revisions and the positive assessments from Reviewers 2 and 3, I believe the manuscript is now suitable for publication.

Reviewer 2 ·

Basic reporting

The author has made as many necessary revisions as possible, and the current version is relatively complete.

Experimental design

no comment

Validity of the findings

no comment

Reviewer 3 ·

Basic reporting

The authors have made appropriate revisions and replies to the comments. No more comments.

Experimental design

no comment

Validity of the findings

no comment